# Affordability of Habitual (Unhealthy) and Recommended (Healthy) Diets in the Illawarra Using the Healthy Diets ASAP Protocol

**DOI:** 10.3390/ijerph22050768

**Published:** 2025-05-13

**Authors:** Kathryn Fishlock, Shauna Gibbons, Karen Walton, Katherine Kent, Meron Lewis, Karen E. Charlton

**Affiliations:** 1School of Medical, Indigenous and Health Sciences, Faculty of Science, Medicine and Health, University of Wollongong, Northfields Ave, Wollongong, NSW 2500, Australia; kjf984@uowmail.edu.au (K.F.); shauna.gibbons@health.nsw.gov.au (S.G.); kwalton@uow.edu.au (K.W.); katherinek@uow.edu.au (K.K.); 2School of Public Health, Faculty of Medicine, The University of Queensland, 288 Herston Road, Herston, QLD 4006, Australia

**Keywords:** diet affordability, food prices, food costs, food insecurity, Healthy Diets ASAP tool, INFORMAS

## Abstract

Amidst a period of sustained inflation and rising living costs, food insecurity is a growing concern in Australia and is correlated with poor diet quality and increased rates of non-communicable diseases. Currently there is a gap in knowledge of the impact of increasing cost-of-living pressures on the affordability of a healthy diet. As affordability plays a key role in food security, this cross-sectional study aimed to examine the costs, affordability, and differential of habitual (unhealthy) and recommended (healthy) diets within the Illawarra region of Australia and compare results to 2022 findings. The Healthy Diets Australian Standardised Affordability and Pricing tool was applied in six locations in the Illawarra, with two randomly selected each from a low, moderate, and high socioeconomically disadvantaged area. Costs were determined for three reference households: a family of four, a single parent family, and a single male. Affordability was determined for the reference households at three levels of income: median gross, minimum-wage, and welfare dependent. Data was compared to data collected in 2022 using the same methods and locations. Recommended diets cost 10.3–36% less than habitual diets depending on household type, but remained unaffordable for welfare dependant households and family households from socioeconomically disadvantaged areas, where diets required 25.5–45.9% of household income. Due to income increases, affordability of both diets has marginally improved since 2022, requiring 0.5–4.8% less household income. This study provides updated evidence that supports the urgent need for policies, interventions, and monitoring to widely assess and improve healthy diet affordability and decrease food insecurity rates. Possible solutions include increasing welfare rates above the poverty line and utilising nudge theory in grocery stores.

## 1. Introduction

Food security is defined as having regular and reliable physical, social, and economic access to enough food, without resorting to charitable food supplies, scavenging, stealing or other coping strategies [1]. Food should be safe to consume, culturally appropriate, and meet all nutritional needs and food preferences [1].

Food insecurity exists on a spectrum, ranging from fear of running out of food (marginal), reduced quality, variety, and desirability of food (moderate), to reduced food intake and regularly skipping meals due to an inability to buy more (severe) [2]. In 2024, 32% of Australians were estimated to be experiencing moderate to severe food insecurity, with an additional 11% facing marginal food insecurity [3]. These devastatingly high rates disproportionally impact vulnerable populations, including lower income, and welfare-dependent households [4,5]. Food insecurity is correlated with diets low in fruits, vegetables, wholegrains, and fibre, and high in saturated fats, sodium, and sugar [6,7,8,9]. This is concerning as poor diets are associated with decreased mental and physical health outcomes and increased rates of preventable non-communicable diseases, which is a major public health issue [8,10].

Food affordability correlates to the economic access component of food security and is the major risk factor in Australia [11]. When a household needs to spend more than 30% of disposable income to purchase adequate food and drinks, this is considered unaffordable [12,13,14]. A range of 25–30% of disposable income required is commonly used as an indicator of food stress and increased vulnerability to food insecurity [12,15]. As cost can substantially influence dietary choices, particularly for those with lower incomes, affordability of a healthy diet has become an important topic of discussion [13,16,17]. Australia does not regularly monitor the cost of eating in line with the Australian Dietary Guidelines (ADG), or the cost differential between a ‘healthy’ and ‘unhealthy’ diet, which is essential for informing and evaluating policies to encourage healthy food consumption [5,13]. Over 60 different surveys or food baskets have been used since 1995 to evaluate food affordability, each with varying methods, resulting in data that is not easily comparable [13]. Many food baskets within these studies did not align with ADG recommendations, replicate typical ‘unhealthy’ purchasing behaviours, or provide cost comparisons between the two, leaving demand for a more rigorous and standardised approach [5,12,18].

The Illawarra Healthy Food Basket (IHFB) survey was implemented biannually between 2000 and 2019, and provided regular comparable data for the Illawarra region, NSW. An analysis of trends between 2011 and 2019 found that affordability remained relatively stable over time for a five-person family on average weekly earnings that included a pensioner aged over 65 years. However, a family of four receiving welfare benefits would have struggled to afford the healthy food basket in 2019, costing 33% of household income [11]. Since 2019, the Healthy Diets Australian Standardised Affordability and Pricing protocol (HD-ASAP) has replaced the IHFB to enable consistency across diet pricing surveys undertaken throughout the rest of Australia, with the previous survey in the Illawarra being conducted in 2022 (unpublished) and reported in the current study for comparative purposes.

The HD-ASAP protocol was developed under the leadership of Lee [5,12] to meet the demand for a more standardised, rigorous, and systematic approach [12,13]. Stemming from the International Network for Food and Obesity/non-communicable Diseases Research, Monitoring and Action Support’s (INFORMAS) “optimal approach” recommendations, application of the protocol allows cost and affordability comparisons between recommended (based on the ADG) and habitual (based on National Nutrition Survey dietary intake data) diets, across time, location, and household types [12].

Given the cost-of-living crisis and increased rates of food insecurity impacting adherence to the ADG [19], it is imperative that the affordability of diets aligning with health recommendations are regularly monitored and reported. As poor diets and consequential health outcomes are a major public health concern, up-to-date data is required to inform targeted policy changes and interventions to reduce food insecurity rates. Therefore, this study aimed to assess the affordability, cost, and differentials between recommended and habitual diets in the Illawarra region for various household types and incomes. It also aimed to compare outcomes to those previously observed in 2022 [20].

## 2. Materials and Methods

This cross-sectional study applied the HD-ASAP protocol to assess cost, affordability, and differential of habitual and recommended diets in the Illawarra region between varying households, incomes, and areas of different socioeconomic status [12]. The protocol consists of five components: a standardised pricing tool for each diet, location sampling and store selection methods, data collection protocols, calculation of household incomes, and analysis and reporting methods. The protocol is described briefly below, and further details, including the decision-making process and testing, are described elsewhere. [12].

The reference household types from the HD-ASAP protocol reported in this study were guided by the most recent Census data and vulnerable groups of concern. For the Illawarra region of NSW, 71% of households were families, averaging two children per family with children, and 25% of households occupied by a single person [21]. Single parent families were included as a vulnerable group of concern, because they occupied larger proportions of households in areas of high socioeconomic disadvantage [Appendix A] [2,22]. The pricing tools were thus applied to assess fortnightly diet costs for three reference households: a family of four (adult male 31–50 years old, adult female 31–50 years old, 14-year-old boy, and 8-year-old girl), a single parent family (adult female 31–50 years old, 14-year-old boy, and 8-year-old girl), and a single male household (adult male 31–50 years old).

### 2.1. Standardised Pricing Tool

As per the HD-ASAP protocol, recommended and habitual diet costs were based on types and quantities of foods and drinks required for each household according to the ADG and as reported in the 2012–2013 National Nutrition Survey (NNS), respectively [Appendix A] [23,24]. The recommended diet contains all the healthy items included in the habitual diet, adjusted to reflect quantities that align with the ADG, and excludes discretionary items [12,23]. The foods and drinks included for each diet are listed in Table 1, and prices were collected for popular branded products, as specified in the protocol.

### 2.2. Sample Location and Store Selection

Statistical Area Level 2 (SA2) locations were stratified by Index of Relative Socioeconomic Disadvantage (IRSD) and assigned quintiles according to the Socio-Economic Indexes for Areas (SEIFA), which are ranked from most disadvantaged (quintile 1) to least disadvantaged (quintile 5), as defined by the Australian Bureau of Statistics (ABS) [12,26]. In 2022, locations were selected according to the HD-ASAP protocol, with each SA2 location within the Illawarra from quintiles 1, 3, and 5 assigned a number and entered into a random number generator to select two sampling locations per quintile, with six locations selected in total. These selected locations were repeated in this study and include Port Kembla-Warrawong and Warilla (Q1); Unanderra-Mount Kembla and Balgownie-Fairy Meadow (Q3); and Thirroul-Austinmer-Coalcliff and Helensburgh (Q5). For each location, the following stores were identified via Google Maps^TM^: two large supermarkets (one from each major chain), an independent grocery store, two major fast food chain outlets (burger and pizza), a local bakery, liquor store, and a fish and chips shop. Stores were ideally located within 7 km of the city centre, with stores selected in 2022 re-surveyed, but if unavailable, a new, comparable store was located [27]. For locations which did not have one of the required stores within a reasonable travelling distance, that store was not included in the study, and an appropriate substitute was selected as per the HD-ASAP protocol.

### 2.3. Data Collection Protocols

Prior to data collection, the single student dietitian collector was trained in the HD-ASAP Protocol [List S1] and given access to the HD-ASAP portal, an online database to support data collection and analysis, housed by the University of Queensland [12,28]. Each store surveyed was assigned a unique reference code, which was entered to gain access to online data entry [Appendix A], with collection undertaken at the same time of year as the previous collection (± one month buffer). All data was collected using non-sales pricing and within a one month period to increase accuracy, as prices are subject to change [12]. Collected data was checked for entry errors by the collector prior to leaving each store and again via the online database upon completion of the data collection phase. Entries were then rechecked for flagged items (item > $30, or brand/size different to protocol). Letters outlining the purpose of the research were provided to each store on the day of collection (22–29 April 2024), and in advance via email for major chain stores, to inform them of the research being conducted and allow for withdrawal from the study prior to collection if desired. For stores that did not consent to participate, the mean price of each food and drink item was obtained from alternative locations (visited or accessed online).

### 2.4. Calculation of Household Incomes

Three income variations were considered in data analysis. The HD-ASAP protocol was applied to calculate median gross, minimum wage, and welfare dependant household incomes [12]. Weekly gross median income (before tax and any expenses) was calculated from median household or personal (single parent or single male households) weekly income for each SA2 location community profile from the August 2021 ABS census data [29], adjusted according to Wage Price Index (WPI) until March 2024 [30], and multiplied by two for fortnightly income [12]. Minimum wage and welfare dependant income did not differ between SEIFA quintiles and were calculated based on minimum wage rates and applicable monetary welfare payments, as per Fair Work Ombudsman and Services Australia, respectively [12,31,32]. A set of assumptions was made for each household type, including the following: rental payments, employment, child vaccination status and school attendance, nil/negligible savings investments, and nil disability payments or child support received [12]. Tax payable was deducted for each household, where applicable, with detailed calculations in Appendix A.

### 2.5. Analysis and Reporting Methods

For each household in each location, the HD-ASAP portal generated an Excel^TM^ spreadsheet providing analysis of total recommended and habitual diet costs and affordability, as well as costs associated with each ADG food group and sub-group [Appendix A; Appendix A] [12,33]. Jamovi^TM^ was used to perform statistical analysis with statistical significance set to *p* ≤ 0.05 [34]. Tests included the following: descriptive statistics to calculate mean cost and standard deviation for each diet according to SEIFA quintile, Shapiro–Wilk to assess data distribution normality, paired *t*-test to compare data from 2022 and 2024, one-way ANOVA-F to compare diet costs and median gross income between quintiles, and independent *t*-tests to compare mean recommended and habitual diet costs for each household. Differentials between the two diet types and affordability thereof were calculated and expressed as a percentage rounded to one decimal place for each diet, household, and income variation. Diets costing more than 30% of household income were deemed unaffordable and representing a high risk of food insecurity, with diets costing 25–30% of income indicative of food stress and increased vulnerability to food insecurity [12,15]. For the purpose of comparison over time, data from 2022 was sourced from a currently unpublished report that utilised the same methods for a family of four on low indicative disposable and median gross incomes [20].

As data was publicly available and did not involve personal information or participant interaction, this project was exempt from ethics committee approval.

## 3. Results

### 3.1. Data Collection

For SEIFA quintiles 1 and 3, data was successfully collected from all stores required, except for two locations, where the independent supermarkets in the area declined to participate. For these stores the mean price from four other stores (i.e., independent supermarkets from the three other locations collected, plus one online location) was used. No online data existed for stores within the same quintile; therefore, online data from stores from varying quintiles was accessed for this purpose. This did not introduce bias since no difference in food costs was seen between quintiles (*p* ≥ 0.98).

For one location in quintile 5, only one large supermarket was in the area, meaning that data from the remaining large and independent stores could not be collected. There were also no fast food chains in that location; therefore, similar foods from local venues were selected as a replacement. These decisions meant that results reflected supermarket and convenient store choices for a household living in that particular area.

### 3.2. Cost of Recommended and Habitual Diets

Recommended diets cost 10.2–36.0% less than habitual diets (Table 2), dependant on the household type, with single male households having the largest difference in cost between the two diets. Diet costs were not significantly different between SEIFA quintiles, so the mean cost for all SEIFA areas combined was used for all calculations.

### 3.3. Affordability of Recommended and Habitual Diets

Neither the recommended nor habitual diet was affordable for all households and income types, with the recommended diet requiring above 25–30% of household income for some families (Table 3). Only single male households, families of four from SEIFA quintiles 3 and 5 based on median gross income, and single parent households on minimum wage required less than 25% of household income to purchase the modelled recommended diet. Families from areas of high socioeconomic disadvantage or those reliant on welfare payments, particularly single parent families, were more likely to require over 25–30% of income to purchase adequate groceries for either diet.

### 3.4. Comparison of Cost and Affordability from 2022 to 2024

Between 2022 and 2024, the recommended diet cost for a family of four within the surveyed region increased by 6.1% ($47), and habitual diet cost increased by 8.8% ($85) (Table 4). Despite the WPI only increasing 7.8% since the last data collection period [30], median household income for the surveyed locations in SEIFA quintiles 1, 3, and 5 increased by 17.3%, 23.5%, and 16.7%, respectively, and minimum wage income for a family of four increased by 13.0%, some of which can be attributed to slight increases in relevant welfare payments [35].

As household incomes increased more than diet costs, the percentage of household income required to purchase each diet has decreased by a mean of 2.8–2.9% for median gross incomes and 1.0–1.7% for minimum wage incomes (Table 5). The largest change occurred for median gross incomes from quintiles 1 (4.0–4.5%) and 3 (3.3–3.5%).

## 4. Discussion

This study is the first to collect data for three household types based on three income variants in the Illawarra region of New South Wales, Australia, and identify how the current cost-of-living crisis, resulting from a period of high and sustained inflation, is impacting the affordability of recommended and habitual diets. Application of the HD-ASAP protocol supports comparison over time and across varying locations within Australia [12].

### 4.1. Comparing Habitual and Recommended Diet Costs

Recommended diets were shown to cost less than habitual diets by 7.9–38.2%, which is consistent with previous studies from varying locations [17,25,35,36,37]. The least difference in cost between diets was for single parent families, followed by families of four, with single male households having the largest gap, with a mean difference of 10.8%, 18.8%, and 36.0%, respectively. The biggest contributor to this difference was the higher mean reported consumption of alcoholic beverages by males compared to females in the dietary intake survey that informs the habitual diet construct [Appendix A; Appendix A]. According to Consumer Price Index (CPI) reports, the cost of foods and non-alcoholic beverages increased by 11.8% from March 2022 to March 2024 [38]. The largest contribution to these changes were for breads and cereal products (19%), food products not elsewhere classified, specifically discretionary items such as chocolate and potato chips (16.5%), meals out and takeaway foods (12.6%), non-alcoholic beverages (10.7%), and fruit and vegetable prices (4.6%), with seasonal variations in prices, as well as the impacts of natural disasters greatly impacting prices in a volatile manner [38,39]. Overall, the cost of healthy foods have increased more than discretionary foods since June 2013, with the trend beginning to level out for some food categories in 2023 [40].

This begs the question, if the recommended diet costs less, why is there such a consistently strong association between poor quality diets and low income? Food purchasing behaviours are a complex issue, with other influencing factors beyond cost including time, skills, nutrition knowledge, community food environment, family preferences, convenience, and available storage and cooking facilities, to name a few [17,41,42]. There is the element of perception of certain foods being too expensive, foods unlikely to spoil, or, particularly for families, sticking to more palatable and familiar foods to prevent food wastage if rejected by their family members [42,43].

A key consideration when addressing diet costs is the frequent, and heavy, price discounts and widespread advertising applied to discretionary food and drink items in comparison to core food groups such as whole grains, produce, lean proteins, and dairy [44]. Multiple studies have shown that discretionary food and drinks are discounted disproportionately to healthier items and are more likely to be placed at key display points throughout the grocery store, which can impact purchasing behaviours [45,46]. Many studies have also found larger price discounts for discretionary items, with only one study seeing no difference between the two categories [44,45,46].

### 4.2. Diet Affordability

Despite being the more cost-effective option, the modelled recommended diets were still not affordable for many households. Families from socioeconomically disadvantaged areas, single parent households, and welfare dependant family households were least likely to be able to afford enough food to meet their family’s requirements. Further, the burden of increased costs of other basic requirements over the previous two years may displace food budgets, reducing affordability even more. Based on the CPI from March 2022 to 2024, rapid cost increases have occurred for rent (13.7% for Sydney), with the average 2–3 bedroom properties costing $650/week [47], electricity (17.4% with rebate), insurance (25.2%), and high interest rates pressuring those with mortgages [38,48].

Households from the most disadvantaged areas had significantly lower incomes, which is likely indicative of the population demographic. Low socioeconomic areas were associated with a higher proportion of retirees, single parent families, persons not in the labour force, and unemployed persons [Appendix A], which impacts the median wage as many are likely welfare dependant. Welfare payments may also be lower than what was calculated for this study due to many factors, such as income/assets tests or ineligibility for rental assistance [49]. As personal income was used for both single male and single parent households to reflect one income, having the median gross income fall between the two welfare incomes would be expected. There is evidence that the supplemental increase in welfare payments provided during the COVID-19 pandemic in 2020 resulted in a rise in affordability of diets and led to a higher consumption of core (healthy) food groups [17,50]. 

### 4.3. Comparison to 2022 Diet Costs and Affordability

Both modelled recommended and habitual diets for a family of four were found to have marginally improved in affordability since 2022, requiring 1–2.9% less household income on average. Even with the reduction, for families on minimum wage or from highly disadvantaged areas, based on median gross income, these results indicate that both diets remain unaffordable.

An improvement in affordability was unexpected given the current cost-of-living crisis and the number of households relying on food relief [3,51]. This result was not consistent for median gross incomes based on ABS statistics [30,38]. The CPI indicates the cost of food increased by 11.8% from March 2022 to March 2024 with wages only increasing by 7.8% over the same period, but there are some underlying factors to consider [30,38]. Firstly, this study used 2021 census data adjusted for WPI increases from 2021 to 2024, whereas the 2022 study used 2016 census data adjusted for WPI increases from 2016 to 2022 as the 2021 census results had not yet been released at the time of analysis. Between the two studies, both of which used the most recent census results adjusted for WPI, median gross household income for the surveyed areas increased by 16.7–23.5%, much higher than the WPI. This may be due to higher response rates during the 2021 census, which increased accuracy, population changes due to COVID travel restrictions, or demographic/income changes within the area [52]. Secondly, since 2022, minimum wage incomes have increased by 13% for a family of four, mostly attributed to welfare payment increases in September 2023 [35]. The last consideration is that broad data such as WPI, CPI, and census data are population measures, and do not accurately reflect food or income at an individual household level [53,54]. The HD-ASAP protocol includes the same foods each time for standardisation and comparability; however, foods used in the protocol are different to those used to measure CPI, and prices may have changed disproportionately to those measured by the latter [55], or have been impacted by factors during the 2022 collection period, particularly for fruit and vegetable costs associated with the flooding in New South Wales and Queensland [25,56].

### 4.4. Limitations

Limitations to this study are outlined in the original HD-ASAP protocol, which includes the limitations of using median gross income rather than disposable income [12]. The habitual diet (unhealthy) pricing tool is also based on reported dietary intakes of the Australian population, while the recommended diet (healthy) is based on the Australian Dietary Guidelines, which references average requirements. These population summary measures may not necessarily represent the food habits or dietary requirements of individual households. The pricing tools assume zero food waste, and the quantities of food items in the tools are assumed to meet nutritional requirements based on estimated moderate activity levels for each member of the household [12]. Comparing two cross-sectional studies taken two years apart is both a strength and a weakness in that it reflects the same time each year, reducing the impact of seasonal variation, but is also only a snapshot and not representative of year-round data. Finally, results are specific to the Illawarra region of New South Wales, Australia, and may not be generalisable to other locations.

### 4.5. Implications for Research and Policy

This study provides further evidence to support broad federal level policies to help Australian households afford sufficient, healthy food such as raising welfare payments above the poverty line in Australia to improve affordability of the recommended diet and reduce the number of households experiencing food insecurity [25,42,50,51].

Mandatory policies impacting the grocery store environment in line with nudge theory should also be considered to support behaviour changes and encourage the shift to a diet aligned with the ADG [43,44,45]. Extensive bans on promotion of discretionary foods are unlikely to be accepted and passed due to industry kickback and influence, but reducing the frequency and magnitude of promotions, and prime display area devoted to discretionary products should be considered [43,44,45,46]. This could be conducted in conjunction with increasing the affordability, promotion, and display of foods better aligned to the Australian Dietary Guidelines [43,44].

## 5. Conclusions

Data pertaining to food costs, affordability, and trends is essential for informing policies and interventions to reduce rates of food insecurity. Whilst affordability is not the only factor, food security cannot be achieved without it, making it one of the biggest contributors. Though the modelled recommended diet was more affordable than the habitual diet for all households, it remained unaffordable for most single parent households, families from lower SEIFA quintiles based on median gross, minimum wage, and welfare dependant incomes, requiring ≥30% of household income. This is of major public health concern and requires urgent attention. Since data collected in 2022, the recommended diet has increased in cost but has become marginally more affordable due to increases in income from wages and welfare benefits that exceeded the increases in diet cost [35]. Regular Australia-wide monitoring of diet affordability using the HD-ASAP protocol should be implemented to inform policies and assess efficacy of any interventions. To improve affordability, promotion, and consumption of diets in line with the Australian Dietary Guidelines, increasing welfare payments above the poverty line should be considered, along with utilisation of in-store techniques, such as nudge theory and discounting policies, to make eating healthier the easier and most cost-effective choice.

## Figures and Tables

**Table 1 ijerph-22-00768-t001:** Foods and drinks included in recommended and habitual diets for the HD-ASAP pricing protocol. Adapted from [25].

Recommended (Healthy) Diet	Habitual (Unhealthy) Diet
Water (bottled) Fruit: apples, bananas, and oranges Vegetables: baked beans (canned), broccoli, carrot, diced tomatoes (canned), four bean mix (canned), frozen mixed vegetables, frozen peas, iceberg lettuce, onion, potatoes, pumpkin, salad vegetables in sandwich, sweetcorn (canned), tomatoes, and white cabbage Grain (cereals): bread in sandwich, cornflakes, dry water crackers, rolled oats, white bread, white pasta, white rice, wholegrain cereal biscuits (Weet-bix^TM^), and wholemeal bread Lean meats and alternatives: beef mince and steak, cooked chicken, eggs, lamb chops, meat in sandwich, peanuts (unsalted), and tuna (canned) Milk, yoghurt, and cheese: cheddar cheese (full fat, reduced fat), milk (full fat, reduced fat), and yoghurt (full fat plain, reduced fat flavoured) Unsaturated oils and spreads: canola (margarine), olive oil, and sunflower oil	Healthy foods and drinks as per the seven food groups in the ‘Recommended diet’ column; in reduced amounts reflecting reported intakes Artificially sweetened beverages Discretionary (unhealthy) food and drinks: ○ Drinks: Sugar sweetened beverages ○ Cereals, snacks, and desserts: chocolate, confectionary, fruit salad (canned in juice), ice cream, mixed nuts (salted), muffin, muesli bar, potato crisps, savoury crackers, and sweet biscuits ○ Processed meats: beef sausages and ham ○ Spreads, sauces, condiments, and ingredients: butter, salad dressing, tomato sauce, and white sugar ○ Convenience meals: chicken soup (canned), frozen fish fillet (crumbed), frozen lasagne, instant noodles, and meat and vegetable stew (canned) ○ Fast food: hamburger, meat pie, pizza, and potato chips/fries ○ Alcohol: beer (full strength), red wine, whisky, and white wine (sparkling)

**Table 2 ijerph-22-00768-t002:** Mean recommended and habitual diet cost and price differential (95% confidence interval) per fortnight for a family of four, single parent family, and single male household for each SEIFA quintile and all locations.

	SEIFA Quintile	Mean Recommended Diet Cost ($AUD)	Mean Habitual Diet Cost ($AUD)	Differential ($AUD)	Differential (%)
Family of four (adult male 31–50-yo, adult female 31–50-yo, 14-yo boy, and 8-yo girl)	1	785 (777–793)	955 (925–985)	170	17.8
	3	781 (769–793)	940 (916–964)	159	16.9
	5	762 (686–838)	975 (963–987)	213	21.8
	Mean total cost	776 (734–818)	957 (921–993)	181 (156–205) *	18.8
Single parent family (adult female 31–50-yo, 14-yo boy, and 8-yo girl)	1	574 (570–578)	632 (606–658)	58	9.2
	3	572 (564–580)	621 (601–641)	49	7.9
	5	558 (502–614)	645 (643–647)	87	13.5
	Mean total cost	569 (539–599)	633 (607–659)	64 (40–88) *	10.2
Single male (adult male 31–50-yo)	1	211 (207–215)	324 (320–328)	113	34.9
	3	209 (207–211)	321 (317–325)	112	34.9
	5	204 (184–224)	330 (322–338)	126	38.2
	Mean total cost	208 (198–218)	325 (315–335)	117 (110–124) *	36.0

* *p* value = ≤ 0.01 from independent *t*-test.

**Table 3 ijerph-22-00768-t003:** Affordability of habitual and recommended diets calculated using median gross (95% confidence interval), low indicative disposable, and welfare dependant incomes for a family of four, single parent, and single male household for each SEIFA quintile per fortnight.

**Habitual Diet**									
**Household Type**	**Family of Four**			**Single Parent**			**Single Male**		
**SEIFA Quintile**	**1**	**3**	**5**	**1**	**3**	**5**	**1**	**3**	**5**
Median gross income ($AUD) *	2501 (2245–2757)	3868 (2990–4746)	5405 (5029–5781)	1238 (1128–1348)	1622 (1514–1730)	2197 (2169–2225)	1238 (1128–1348)	1622 (1514–1730)	2197 (2169–2225)
Affordability (%)	38.3	24.7	17.7	51.1	39.0	28.8	26.2	20.0	14.8
Minimum wage ($AUD)	3041			2437			1535		
Affordability (%)	31.5			26.0			21.2		
Welfare dependant income ($AUD)	2244			1923			887		
Affordability (%)	42.6			32.9			36.6		
**Recommended Diet**									
**Household Type**	**Family of Four**			**Single Parent**			**Single Male**		
**SEIFA Quintile**	**1**	**3**	**5**	**1**	**3**	**5**	**1**	**3**	**5**
Median gross income ($AUD) *	2501 (2245–2757)	3868 (2990–4746)	5405 (5029–5781)	1238 (1128–1348)	1622 (1514–1730)	2197 (2169–2225)	1238 (1128–1348)	1622 (1514–1730)	2197 (2169–2225)
Affordability (%)	31	20.0	14.4	45.9	35.0	25.8	16.8	12.8	9.5
Minimum wage income ($AUD)	3041			2437			1535		
Affordability (%)	25.5			23.3			13.5		
Welfare dependant income ($AUD)	2244			1923			887		
Affordability (%)	34.6			29.5			23.4		

* *p* value = ≤ 0.05 from one-way ANOVA-F between quintiles. Orange = ≥ 30% of income required and not affordable; yellow = 25–30% of income required, indicative of food stress.

**Table 4 ijerph-22-00768-t004:** Mean cost and differential (95% confidence interval) per fortnight for habitual and recommended diets in 2022 and in 2024 for a family of four.

	SEIFA Quintile	Mean Cost 2022 ($AUD)	Mean Cost 2024 ($AUD)	Differential ($AUD)	Differential (%)
Recommended diet	1	725 (717–733)	785 (777–793)	60	7.6
	3	745 (721–769)	781 (769–793)	36	4.6
	5	719 (653–785)	762 (686–838)	43	5.6
	Mean total	729 (689–769)	776 (734–818)	47 (32–62) *	6.1
Habitual diet	1	877 (863–891)	955 (925–985)	78	8.2
	3	880 (876–884)	940 (916–964)	60	6.4
	5	859 (833–885)	975 (963–987)	116	11.9
	Mean total	872 (848–896)	957 (921–993)	85 (56–114) *	8.8

Note: ($AUD) Results are rounded to the nearest whole number, and (%) results to the nearest single decimal place. * *p* = ≤ 0.01 from paired *t*-test.

**Table 5 ijerph-22-00768-t005:** Proportion of household income (affordability) required to purchase habitual and recommended diets for a family of four on minimum wage and median gross incomes and differential between 2022 and 2024.

	SEIFA Quintile	Median Gross Income			Minimum Wage Income		
		Mean Affordability 2022 (%)	Mean Affordability 2024 (%)	Differential (%)	Mean Affordability 2022 (%)	Mean Affordability 2024 (%)	Differential (%)
Recommended diet	1	35.0	31.0	−4.0	27.0	25.5	−1.5
	3	23.5	20.0	−3.5	27.5	25.5	−2.0
	5	15.5	14.4	−1.1	27.0	25.5	−1.5
	Mean total affordability	24.7	21.8	−2.9	27.2	25.5	−1.7
Habitual diet	1	42.5	38.3	−4.2	32.5	31.5	−1.0
	3	28.0	24.7	−3.3	33.0	31.5	−1.5
	5	18.5	17.7	−0.8	32.0	31.5	−0.5
	Mean total affordability	29.7	26.9	−2.8	32.5	31.5	−1.0

## Data Availability

For privacy reasons, data are stored on a secure server and is not publicly available. Data may be provided upon application to the authors.

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
