# Peer review of "Affordability of Habitual (Unhealthy) and Recommended (Healthy) Diets in the Illawarra Using the Healthy Diets ASAP Protocol"

_ijerph, 2025, doi:10.3390/ijerph22050768_

Round 1
Reviewer 1 Report
Comments and Suggestions for Authors
Thank you for a well written manuscript regarding a significantly important issue across Australia, namely, (in)affordability of a healthy diet.
Abstract:
line 9 - "highly" should be 'high'
line 12 - income: median gross
Introduction:
This is well structured, with a strong narrative regarding food security and food affordability.
The rationale for the choice of the ASAP tool is well substantiated - consider including this more recent sys review regarding food costing methods and tools [Russell C, Whelan J, Love P. Assessing the Cost of Healthy and Unhealthy Diets: A Systematic Review of Methods. Curr Nutr Rep. 2022 Dec;11(4):600-617. doi: 10.1007/s13668-022-00428-x. Epub 2022 Sep 9. PMID: 36083573; PMCID: PMC9461400.]
Methods:
A comprehensive overview is provided.
2.3 Data collection
- I suggest reordering the flow here - a) training of data collector b) approaching and gaining consent of stores c) data collection and checking of errors
- How store owners were contacted and permission obtained might fit better here (rather than 2.5) How did stores contact you, and by when, if choosing to withdraw?
- Checking of errors needs further clarification about who did this and how.
- The section "Three independent .... (P=>0.98)." fits under results
Results:
It would be useful to start with an overview of the final number and type of stores included in the study, and any online ie: did you reach your targets for each location within Q1, Q3 and Q5 for 2 large supermarkets (one from each major chain), 1 independent grocery store, 2 major fast food chain outlets (burger and pizza), 1 local bakery, 1 liquor store and 1 fish and chip shop?
This is where you can then describe:
- Helensburgh (only 1 large supermarket); it is also unclear why you replaced fast food chain food items in this area? [rather than currently in 2.2]
- 3 independent grocers not consenting [2.3] - it's unclear which Qs these represent? why the decision to then gather data from 4 other stores - and of these how many were visited and how many were online data?
Cost and affordability results and comparison across the two surveys are well described with accompanying tables.
Discussion:
Your main messages, from your abstract, would appear to be 1) solutions to increase welfare rates above the poverty line, and b) utilising nudge theory in grocery stores.
Your discussion could be strengthened by focusing on these proposed solutions, providing examples of policy/program intervention implemented elsewhere that have aimed to, and potentially achieved, improvements in healthier diet choices - for example, during COVID when welfare payments increased, and work done by Cameron et al within Australian supermarkets. While this is described briefly in section 4.5, integrating the proposed solutions into each section would be more impactful.
Section 4.1 - Reference 36 is the Food Environment Dashboard, which is a collation of relevant data, however it is recommended that you include the actual references here for these specific studies - Herath 2023, Lee, Patay 2021, Lee, Kane 2020, Summons 2020, Riesenberg 2019, Lee, Lewis 2018, Love 2018
Herath MP, Murray S, Lewis M, Holloway TP, Hughes R, et al - doi: 10.3390/nu15183908
Lee, A., Patay, D., Herron, LM. - https://doi.org/10.1186/s12939-021-01481-8
Lee, A.J., Kane, S., Herron, LM.- https://doi.org/10.1186/s12966-020-00981-0
Lee & Lewis - https://doi.org/10.3390/ijerph15122912
Love, Whelan, Bell et al - https://doi.org/10.3390/ijerph15112469
Riesenberg D, Backholer K, Zorbas C, Sacks,G, et al - https://doi.org/10.2105/AJPH.2019.305229
Summons et al - Cost-and-affordability-of-healthy-sustainable-and-equitable-diets-in-the-Torres-Strait-Islands.pdf
Note that this is not a peer-reviewed manuscript – it is a summary results brief – so probably shouldn’t be included as a primary reference.
Conclusion:
This appears somewhat repetitive of the results and could be condensed to the key points that are also of relevance to other parts of Australia ie: modelled healthy diets cost less than current unhealthy diets but are still unaffordable for most households despite income increases over time; we need regular monitoring to inform system level changes (welfare payments) and local-level strategies that can support retail outlets to improve promotion and affordability of healthier food choices.
References:
Ref 36 - see comment above
Ref 54 - This is a media piece - suggest citing Christina Zorba's original work here
Minor grammar errors:
Section 4.1 - paragraph 1
- check spacing between words from "According to Consumer.... food categories in 2023."
- last sentence - "Overall, the cost of healthy foods has increased...
Section 4.1 - last paragraph - do you mean 'fresh produce' specifically fruit and veg?
Section 4.4 - spelling of "Comparing" (2nd last sentence)
Reviewer 2 Report
Comments and Suggestions for Authors
The manuscript describes the comparison of costs, affordability of habitual vs. healthy diets in hypothetical reference households within the Illawarra region of Australia. The data was also compared with available data of 2022. The results show that healthy diets costs 10-36% less than the habitual diets but were still unaffordable for welfare recipients and households from socio-economically disadvantaged areas.
This is a very important and well written manuscript.
The reviewer only has minor comments which mainly relate to the clarity of certain aspects. The abstract uses the wording “habitual (unhealthy) and recommended (healthy) diets”, the heading of table 1 refers to “current and recommended diets”. It would help the reader to stay consistent with the wording.
Page 4, 2.2 Should it not be fish and chipS shop?
Page 5, 2.3 Please add a sentence on how sale pricings were handled.
Page 7, 3.2 typing error: …on minimum wage…
Page 10, 4.1 Consider storage possibilities as another influencing factor. Storage space has shown to be critical for families at risk of poverty and food bank users. But research also shows that certain “healthier” foods need to stored in cold areas or elsewhere and are only more affordable if bought in larger quantities.
Reviewer 3 Report
Comments and Suggestions for Authors
Interesting article that highlights the cost of a recommended adequate diet compared to the usual diet. I suggest the authors describe the subsidies that low-income families or individuals receive, although these are social policy actions in developed countries. Readers from developing countries might misinterpret or not know whether they are monetary or food subsidies.
I also suggest including among the factors that can influence the purchase of regular foods instead of recommended ones, despite a lower price, the impact of food advertising, especially for processed foods, which can be relatively less expensive and highly palatable.It would also be appropriate to indicate how poverty is measured in Australia: whether it is based solely on the indicator of the percentage of the household budget spent on food purchases, or whether there are other methods that relate to the poverty line based on the usual consumption basket. This could be described either in the introduction or in the discussion
Reviewer 4 Report
Comments and Suggestions for Authors
This study examines the affordability of healthy vs unhealthy diets in the Illawarra region of Australia. This research highlights that healthy diets are generally cheaper than unhealthy ones, but they remain unaffordable for welfare-dependent households. The study calls for urgent policy action to address food insecurity. However, the style of the abstract should be changed and more details on data analysis is needed, especially the impact of inflation.
The introduction mentions the limitations of previous studies but doesn't explain how the HD-ASAP protocol overcomes these issues. A clearer comparison between past methods and the new approach would improve the quality of the introduction.
While the study emphasizes the importance of food affordability monitoring, it lacks a detailed discussion of policy applications.
The paper focuses on the Illawarra region but doesn’t address how these findings apply to other regions. There is a need to introduce the broader applicability of the results.
The authors should provide more information on how their study addresses the food affordability issues faced by vulnerable groups.
To improve the section of the Materials and Methods, please clarify the store and location selection process, including how socioeconomic factors influenced the sampling. Provide more details on the pricing tools used, statistical methods. Please include handling of missing data and potential biases to ensure transparency of the methodology.
The study relies on median gross income to assess affordability, which may not accurately reflect the actual disposable income of households.
The limitations of the paper include the use of gross income instead of disposable income, regional specificity, the assumption of zero food waste, and the reliance on broad population data rather than individual household dietary needs. The authors should include the limitations of the study.
Comments on the Quality of English LanguageMinor editing is needed.
Round 2
Reviewer 4 Report
Comments and Suggestions for Authors
The authors made significant changes and the paper can be published in a current form.
Comments on the Quality of English LanguageMinor editing is needed.